# Inherited Retinal Degeneration: PARP-Dependent Activation of Calpain Requires CNG Channel Activity

**DOI:** 10.3390/biom12030455

**Published:** 2022-03-15

**Authors:** Jie Yan, Alexander Günter, Soumyaparna Das, Regine Mühlfriedel, Stylianos Michalakis, Kangwei Jiao, Mathias W. Seeliger, François Paquet-Durand

**Affiliations:** 1Cell Death Mechanism Group, Institute for Ophthalmic Research, University of Tübingen, 72076 Tübingen, Germany; jieyan19910809@hotmail.com (J.Y.); soumyaparnadas@gmail.com (S.D.); 2Graduate Training Centre of Neuroscience, University of Tübingen, 72076 Tübingen, Germany; 3Division of Ocular Neurodegeneration, Institute for Ophthalmic Research, University of Tübingen, 72076 Tübingen, Germany; alexander.guenter@uni-tuebingen.de (A.G.); regine.muehlfriedel@med.uni-tuebingen.de (R.M.); 4Department of Ophthalmology, University Hospital, LMU Munich, 80539 München, Germany; michalakis@lmu.de; 5Key Laboratory of Yunnan Province, Affiliated Hospital of Yunnan University, Kunming 650051, China; kangwei.jiao@ynu.edu.cn

**Keywords:** retinitis pigmentosa, calcium, cGMP, nonapoptotic cell death, PKG, HDAC, photoreceptor degeneration

## Abstract

Inherited retinal degenerations (IRDs) are a group of blinding diseases, typically involving a progressive loss of photoreceptors. The IRD pathology is often based on an accumulation of cGMP in photoreceptors and associated with the excessive activation of calpain and poly (ADP-ribose) polymerase (PARP). Inhibitors of calpain or PARP have shown promise in preventing photoreceptor cell death, yet the relationship between these enzymes remains unclear. To explore this further, organotypic retinal explant cultures derived from wild-type and IRD-mutant mice were treated with inhibitors specific for calpain, PARP, and voltage-gated Ca^2+^ channels (VGCCs). The outcomes were assessed using in situ activity assays for calpain and PARP and immunostaining for activated calpain-2, poly (ADP-ribose), and cGMP, as well as the TUNEL assay for cell death detection. The IRD models included the *Pde6b-*mutant *rd1* mouse and *rd1*Cngb1^−/−^* double-mutant mice, which lack the beta subunit of the rod cyclic nucleotide-gated (CNG) channel and are partially protected from *rd1* degeneration. We confirmed that an inhibition of either calpain or PARP reduces photoreceptor cell death in *rd1* retina. However, while the activity of calpain was decreased by the inhibition of PARP, calpain inhibition did not alter the PARP activity. A combination treatment with calpain and PARP inhibitors did not synergistically reduce cell death. In the slow degeneration of *rd1***Cngb1^−/−^* double mutant, VGCC inhibition delayed photoreceptor cell death, while PARP inhibition did not. Our results indicate that PARP acts upstream of calpain and that both are part of the same degenerative pathway in *Pde6b*-dependent photoreceptor degeneration. While PARP activation may be associated with CNG channel activity, calpain activation is linked to VGCC opening. Overall, our data highlights PARP as a target for therapeutic interventions in IRD-type diseases.

## 1. Introduction

Inherited retinal degenerations (IRDs) are a genetically diverse group of diseases that typically result in progressive photoreceptor cell death, severe visual handicap, and blindness [1]. The most common disease within the IRD group is retinitis pigmentosa (RP) [2], in which patients initially experience night blindness and gradual constriction of the visual field due to primary loss of rod photoreceptors. This is followed by secondary degeneration of cone photoreceptors, eventually resulting in complete blindness [3]. Approximately one in four thousand people are affected by RP [2]. IRD-type blinding diseases are generally considered to be untreatable [4]. The second messenger cyclic-guanosine-monophosphate (cGMP) has been found to play a central role in the pathobiology of many genetically distinct types of IRD [5], and excessive cGMP-signaling may be directly or indirectly associated with the activity of poly (ADP-ribose) polymerase (PARP), cyclic nucleotide-gated (CNG) channels, and calpain-type proteases [5,6,7].

One of the best studied animal models for IRD is the *rd1* mouse (retinal degeneration 1), a naturally occurring mouse model first described by Keeler in the early 1920s [8]. In *rd1* mice, the gene encoding for the beta subunit of the rod photoreceptor-specific phosphodiesterase-6 (PDE6) is mutated [9], causing PDE6 dysfunction, accumulation of cGMP in rod photoreceptors, and primary rod cell death, followed by secondary cone photoreceptor cell loss [10,11]. The degeneration of rods in the *rd1* mouse is associated with a prominent activation of both PARP and calpain [12,13]. In humans, 4 to 5% of IRD patients carry mutations in *PDE6* genes, making it seem likely that they will also suffer from high cGMP levels and the corresponding up-regulation of downstream cGMP-signaling targets [14].

PARP is a DNA repair enzyme [15] and catalyzes ADP-ribose transfer to target proteins [16]. It can sequentially add ADP-ribose units from nicotinamide adenine dinucleotide (NAD^+^) to form polymeric ADP-ribose (PAR) chains [17]. The enzymatic activity of PARP has been related to a variety of different cellular functions, including DNA repair and transcription, regulation of gene expression, metabolism, and aging [18]. However, PARP may also be the primary driver for a specific form of cell death, termed PARthanatos [19]. The current view is that in IRD nonapoptotic cGMP-dependent cell death is characterized by PARP over-activation and the accumulation of PAR [14], indicating a possible crosstalk between cGMP signaling and PARthanatos [6].

Calpain is a Ca^2+^-dependent thiol protease, which has been implicated in fundamental cellular processes, including cell proliferation, apoptosis, and differentiation [20]. The Ca^2+^ influx required for calpain activation may occur via CNG channels located in the plasma membrane of the photoreceptor outer segment. These channels are gated by cGMP, which is often elevated in retinal degeneration [7,21,22]. The activation of CNG channels leads to a depolarization of the photoreceptor plasma membrane, which activates synaptic voltage-gated Ca^2+^ channels (VGCCs), increasing the synaptic and cytosolic Ca^2+^ levels [23]. The influx of Ca^2+^ via CNG channels and/or VGCC is thought to cause the activation of calpain-type proteases, which may promote cell destruction [24,25,26,27].

In IRD, PARP and calpain have been suggested to be part of two independent cell death subroutines, both of which are triggered by elevated levels of cGMP [5]. On the other hand, PARP is known to be cleaved by calpain, indicating that calpain could potentially control PARP activity [28,29]. Conversely, in a model for NMDA toxicity in rat primary cortical neurons, PARP was found to regulate the calpain activity via mitochondrial Ca^2+^ homeostasis [30]. We hypothesized that PARP and calpain crosstalk could also occur during cGMP-dependent cell death in IRD. To investigate this possibility, we used specific inhibitors for PARP, calpain, and VGCC to block the corresponding downstream pathways and assessed the contribution of the rod CNG channel via knockout of its beta subunit. Through these interventions, we show that (1) calpain and PARP are part of the same degenerative pathway triggered by high levels of cGMP in photoreceptors and that (2) PARP controls calpain activity, likely indirectly via CNG channel activity and excessive demands on energy metabolism.

## 2. Materials and Methods

### 2.1. Animals

For retinal explant cultures C3H/HeA *Pde6b ^rd1^*^/*rd1*^ animals (*rd1*), their congenic wild-type C3H/HeA *Pde6b*^+/+^ counterparts (*wt*) [31], and B6.129SvJ;C3H/HeA-*CNGB1*^tm^ double-mutant mice (*rd1*Cngb1^−/−^*) were used [22].The *rd1*Cngb1^−/−^* double mutants were generated by an intercross of *rd1* and *Cngb1*^−/−^. Animals were used regardless of gender. The stock has been maintained by repeated backcrossing over 10 generations to make a congenic inbred strain, homozygous for both gene mutations. Animals were housed under standard white cyclic lighting and had free access to food and water. Animal protocols compliant with §4 of the German law of animal protection were reviewed and approved by the Tübingen University committee on animal protection (Einrichtung für Tierschutz, Tierärztlicher Dienst und Labortierkunde, Registration No. AK02/19M).

### 2.2. Retinal Explant Culture

To assess the effects of Olaparib, calpastatin, and D-cis-diltiazem on calpain activity, activated calpain-2, PARP activity, PAR, and photoreceptor degeneration, *rd1* and *rd1*Cngb1^−/−^* retinas were explanted at postnatal day 5 (P5). The explants were cultured on a polycarbonate membrane (Corning-Costar Transwell permeable support, 24-mm insert, #CLS3412) with complete medium (Gibco R16 medium with supplements) [32]. The R16 medium was changed every two days with treatment at either P7 and P9 for *rd1* or at P7, P9, P11, P13, and P15 for *rd1***Cngb1^−/−^* explants. Except for the *wt* situation, the two retinas obtained from a single animal were split across different experimental groups so as to maximize the number of independent observations acquired per animal. The cultures were treated with 20-µM calpastatin, 1-µM Olaparib, 100-µM D-cis-diltiazem, and 20-µM calpastatin combined with 1-µM Olaparib, respectively. In these treatments, Olaparib was dissolved in DMSO at a final medium concentration of 0.1% DMSO. Cultures were ended on P11 (*rd1*) and P17 (*rd1***Cngb1^−/−^*) by either fixation with 4% paraformaldehyde (PFA) or without fixation and direct freezing on liquid N_2_. The explants were embedded in Tissue-Tek (Sakura Finetek Europe B.V., Alphen aan den Rijn, The Netherlands) and sectioned (12 µm) in a cryostat (Thermo Fisher Scientific, CryoStar NX50 OVP, Runcorn, UK).

### 2.3. TUNEL Staining

The TUNEL (terminal deoxynucleotidyl transferase dUTP nick end labeling) assay kit (Roche Diagnostics, Mannheim, Germany) labeled dying cells. Histological sections from retinal explants were dried and stored at −20 °C. The sections were rehydrated with phosphate-buffered saline (PBS; 0.1 M) and incubated with proteinase K (1.5 µg/µL) diluted in 50-mM TRIS-buffered saline (TBS; 1-µL enzyme in 7-mL TBS) for 5 min. This was followed by 3 times 5-min TBS washing and incubation with blocking solution (10% normal goat serum, 1% bovine serum albumin, and 1% fish gelatin in phosphate-buffered saline with 0.03% Tween-20). TUNEL staining solution was prepared using 10 parts of blocking solution, 9 parts of TUNEL labeling solution, and 1 part of TUNEL enzyme. After blocking, the sections were incubated with TUNEL staining solution overnight at 4°C. Finally, the sections were washed 2 times with PB, mounted using Vectashield with DAPI (Vector Laboratories Inc., Burlingame, CA, USA), and imaged under a Zeiss (ApoTome 2) microscope for further analysis.

### 2.4. Calpain-Activity Assay

This assay allows resolving the overall calpain activity in situ on unfixed tissue sections. Retinal tissue sections were incubated and rehydrated for 15 min in a calpain reaction buffer (CRB) (5.96-g HEPES, 4.85-g KCl, 0.47-g MgCl_2_, and 0.22-g CaCl_2_ in 100-mL ddH_2_O; pH 7.2) with 2-mM dithiothreitol (DTT). The tissue sections were incubated for 2.5 h at 37 °C in CRB with tBOC-Leu-Met-CMAC (25-µM; Thermo Fisher Scientific, A6520). Then, the section was washed with PBS and incubated with ToPro (1:1000 in PBS, Thermo Fisher Scientific, OR, USA) for 15 min. Afterwards, the tissue sections were washed twice in PBS (5 min) and mounted using Vectashield without DAPI (Vector Laboratories Inc., Burlingame, CA, USA) for immediate visualization under the ZEISS ApoTome 2.

### 2.5. PARP Activity Assay

This assay allows resolving the overall PARP activity in situ on unfixed tissue sections [33]. Retinal tissue sections were incubated and rehydrated for 10 min in PBS. The reaction mixture (10-mM MgCl_2_, 1-mM dithiothreitol, and 50-μM 6-Fluo-10-NAD^+^ (Biolog, Cat. Nr.: N 023) in 100-mM Tris buffer with 0.2% Triton X100, pH 8.0) was applied to the sections for 3 h at 37 °C. After three 5-min washes in PBS, the sections were mounted in Vectashield with DAPI (Vector Laboratories Inc., Burlingame, CA, USA) for immediate visualization under the ZEISS ApoTome 2.

### 2.6. PAR DAB Staining

For the detection of PAR DAB staining, we used fixed sections. 3,3′-diaminobenzidine (DAB) staining commenced with the quenching of endogenous peroxidase activity using 40% MeOH and 10% H_2_O_2_ in PBS with 0.3% Triton X-100 (PBST) for 20 min. The sections were further incubated with 10% normal goat serum (NGS) in PBST for 30 min, followed by anti-PAR antibody (1:200; Enzo Life Sciences, Farmingdale, NY, USA) incubation overnight at 4 °C. Incubation with the biotinylated secondary antibody (1:150, Vector in 5% NGS in PBST) for 1 h was followed by the Vector ABC Kit (Vector Laboratories, solution A and solution B in PBS, 1:150 each) for 1 h. DAB staining solution (0.05-mg/mL NH_4_Cl, 200-mg/mL glucose, 0.8-mg/mL nickel ammonium sulphate, 1-mg/mL DAB, and 0.1 vol. % glucose oxidase in phosphate buffer) was applied evenly, incubated for precisely 3 min, and immediately rinsed with phosphate buffer to stop the reaction. The sections were mounted in Aquatex (Merck, Darmstadt, Germany).

### 2.7. Calpain-2/cGMP Immunohistochemistry

Sections were rehydrated with PBS for 15 min. The sections were then incubated with blocking solution (10% NGS, 1% BSA 911, and 0.3% PBST) for 1 h. The primary antibodies anti-calpain-2 (ab39165; 1:200; Abcam, Cambridge, UK) and cGMP (1:250; kindly provided by Harry Steinbusch, Maastricht University, Maastricht, The Netherlands) were diluted in blocking solution and incubated overnight at 4 °C. Rinsing with PBS for 3 times 10-min each was followed by incubation with the secondary antibody (Molecular Probes, AlexaFluor568 (A11036), diluted 1:300 in PBS) for 1 h. The sections were further rinsed with PBS for 3 times 10-min each and mounted with Vectashield with DAPI (Vector Laboratories Inc., Burlingame, CA, USA).

### 2.8. Microscopy and Image Analysis in Retinal Cultures

The images of organotypic explant cultures were captured using a Zeiss Imager Z.2 fluorescence microscope, equipped with ApoTome 2, an Axiocam 506 mono camera, and HXP-120V fluorescent lamp (Carl Zeiss Microscopy, Oberkochen, Germany). The excitation (λExc.)/emission (λEm.) characteristics of the filter sets used for the different fluorophores were as follows (in nm): DAPI (λExc. = 369 nm, λEm = 465 nm), AF488 (λExc. = 490 nm, λEm = 525 nm), AF568 (λExc. = 578 nm, λEm = 602 nm), and ToPro (λExc. = 642 nm, λEm = 661 nm). The Zen 2.3 blue edition software (Zeiss) captured images (tiled and z-stack, 20× magnification). Sections of 12-µm thickness were analyzed using 12–15 Apotome Z-planes. For the quantification of positive cells in the retinal ONL, we proceeded as follows: The number of cells in six different rectangular ONL areas was counted manually based on the number of DAPI-stained nuclei and used to calculate an average ONL cell size. This average ONL cell size was then used to rapidly calculate the total number of cells in a given ONL area. The percentage of positive cells was calculated by dividing the absolute number of positive cells by the total number of ONL cells.

### 2.9. Statistical Analysis

Quantitative data was compared by the Student’s *t*-test or Mann–Whitney *U* test. Multiple comparisons were made using the Kruskal–Wallis one-way analysis of variance test. All calculations were performed with GraphPad Prism 8 (GraphPad Software, La Jolla, CA, USA); *p* < 0.05 was considered significant. The figures were prepared using Photoshop CS5 (Adobe, San Jose, CA, USA). The diagram was created with BioRender.com.

## 3. Results

### 3.1. Calpastatin, D-cis-diltiazem, and Olaparib Reduce Calpain Activity in Photoreceptors

To investigate whether and how PARP and calpain could interact with each other, we treated organotypic retinal explants with well-validated and highly selective inhibitors for calpain (i.e., calpastatin) [34] and PARP (i.e., Olaparib) [35]. In addition, we used D-cis-diltiazem to block L-type voltage-gated Ca^2+^ channels (VGCCs) [36]. As readouts, we used in situ activity assays for calpain and PARP, immunolabeling for activated calpain-2 and poly (ADP-ribose) (PAR), as well as the TUNEL assay, to detect cell death. Retinal explant cultures were derived from *wt* and *rd1* animals, explanted at postnatal day 5 (P5), and treated from P7 to P11.

In the *wt* retina, calpain activity and calpain-2 activation were generally relatively low when compared with *rd1*, in which both markers labeled large numbers of positive cells in the ONL (Figure 1a and Appendix A Figure A1a). In *rd1*, treatment with calpastatin and D-cis-diltiazem, as expected, reduced both the calpain activity and calpain-2 activation (*p* < 0.0001, Figure 1d and Appendix A Figure A1d). The solvent DMSO used for Olaparib dissolution did not affect the calpain activity (Appendix A Figure A2a–c). Surprisingly, when the retinal explants were treated with Olaparib, the calpain activity and calpain-2 activation in the ONL was also significantly decreased (*p* = 0.0012, Figure 1e,g and Appendix A Figure A1e,k). However, the combined treatment with calpastatin and Olaparib did not reduce cell death any further compared to the calpastatin or Olaparib single treatments (Figure 1f).

### 3.2. PARP Activity in rd1 Photoreceptors Is Reduced by Olaparib and D-cis-diltiazem but Not by Calpastatin

The reduction of calpain activity after Olaparib treatment suggested that PARP activity controlled calpain activation. To further investigate the effects of PARP activity on calpain, we assessed PARP activity on unfixed retinal tissue sections [33] and stained PFA-fixed retinal tissues for poly (ADP-ribose) (PAR) to evaluate PARP activity also indirectly [13]. Calpastatin, D-cis-diltiazem, Olaparib, and calpastatin combined with Olaparib were used to treat organotypic retinal explant cultures derived from *rd1 mice*. Untreated explant cultures from *wt* and *rd1* animals were used as the control.

In untreated *wt* retina the numbers of photoreceptors in the ONL displaying PARP activity were very low when compared to untreated *rd1* retina (Figure 2a,b). Calpastatin did not reduce the numbers of ONL cells positive for PARP activity or PAR (Figure 2c, quantification in Figure 2g, and Appendix A Figure A1h,l) when compared to the *rd1* untreated control. Interestingly, D-cis-diltiazem significantly reduced the PARP activity (*p* < 0.001, Figure 2d,g) and PAR generation (*p* < 0.01, Appendix A Figure A1i,l) in *rd1* organotypic retinal explant cultures. As expected, Olaparib significantly decreased the PARP activity (*p* < 0.0001, Figure 2e,g) and PAR-positive cells (*p* < 0.05, Appendix A Figure A1j,l), while its solvent DMSO had no effect (Appendix A Figure A2d–f). However, the combination treatment with calpastatin and Olaparib did not reduce the PARP activity further (Figure 2f,g) compared with the Olaparib single treatment.

Together, our data up to this point (Section 3.1 and Section 3.2) indicated that calpain activity was regulated by PARP and that PARP activity in turn might be controlled by VGCC.

### 3.3. rd1 Photoreceptor Degeneration Is Delayed by Calpastatin and Olaparib but Not by D-cis-diltiazem

To further investigate the effects of calpain, VGCC, and PARP inhibition on the progression of *rd1* photoreceptor cell death, we used the TUNEL assay to quantify the numbers of dying cells in the ONL. Since DMSO was used as a solvent for Olaparib, we also tested for the effects of DMSO alone on cell death and found that DMSO had no significant effect (Appendix A
Figure A2g–i).

In *wt* retinal explants, a relatively low number of ONL cells (i.e., photoreceptors) were positive for the TUNEL assay compared with their *rd1* counterparts (Figure 3a,b). Calpastatin treatment (Figure 3c) led to a significant reduction of cell death in *rd1* retina (calpastatin: *p* = 0.0057, quantification in Figure 3g) when compared to the untreated *rd1* control. In contrast, D-cis-diltiazem treatment did not reduce the number of TUNEL-positive dying cells in *rd1* ONL (Figure 3d). Yet, the Olaparib treatment did result in a significant reduction of ONL cell death (Olaparib: *p* < 0.0001, Figure 3e,g). The combination treatment with calpastatin and Olaparib reduced cell death in the ONL (calpastatin and Olaparib: *p* < 0.0001, Figure 3f,g), albeit without showing an additional effect compared with either of the two compounds applied alone. These results provided a strong indication that PARP and calpain were part of the same photoreceptor degenerative pathway.

As an additional control, to rule out possible off-target effects of notably D-cis-diltiazem or Olaparib on cGMP synthesis in photoreceptors, we performed an immunostaining for cGMP on *rd1* retinal explant cultures. The cGMP staining and its quantification did not indicate any treatment-related alterations in cGMP accumulation in ONL cells (Appendix A
Figure A3).

### 3.4. Calpain-2 Activation Is Controlled by Both CNG Channel and VGCC Activity

To further investigate the role of the CNG channel for the activation of calpain and PARP, we used *rd1*Cngb1^−/−^* mice, i.e., *rd1* mice, in which the rod photoreceptor CNG channel was not functional. These animals were generated by crossbreeding the *Pde6b* mutant *rd1* mice with mice that lack the gene encoding for the beta subunit of the rod CNG channel [22]. Without the *Cngb1* subunit rod, CNG channels are not properly formed, and rods essentially lose their ability to generate cGMP-gated currents and responses to light [37]. Previously, we showed that the photoreceptors of *rd1*Cngb1^−/−^* mice are partially protected from cell death, with the peak of cell death shifting from P13 in *rd1* retinas to approximately P18 in *rd1*Cngb1^−/−^* retinas [22]. To account for this slower retinal degeneration phenotype in the double-mutant retina, we performed D-cis-diltiazem and Olaparib treatments on organotypic retinal tissue cultures derived from *rd1*Cngb1^−/−^* animals at P7 and cultured until P17. To assess how the inhibitors affected calpain, we used a calpain activity assay and immunostaining for activated calpain-2.

The double-mutant *rd1*Cngb1^−/−^* mice had a significantly lower number of calpain-positive cells in the ONL than *rd1* mice (Figure 4a–d, *cf*. Figure 1). However, even in the absence of functional rod CNG channels, there were still more calpain-positive cells in the ONL of the *rd1*Cngb1^−/−^* retina than in the *wt* retina (Figure 4a–d). While treatment with D-cis-diltiazem did not decrease the overall calpain activity in double-mutant retina (Figure 4e), it did significantly reduce calpain-2 activation (*p* = 0.0026, Figure 4f) when compared to the *rd1*Cngb1^−/−^* untreated control. As opposed to the *rd1* single mutant situation, in the double-mutant retina, Olaparib failed to reduce the calpain activity and calpain-2 activation. Overall, this data suggested that CNG channel function was required for the PARP-mediated activation of calpain, while calpain-2 activation was dependent on VGCC activity.

### 3.5. D-cis-diltiazem Reduced PARP Activity in the rd1*Cngb1^−/−^ Retina

To dissect the contribution of the CNG channel and the VGCC to the activation of PARP in degenerating *rd1* photoreceptors, we quantified the PARP activity on unfixed retinal tissue sections and assessed the PAR accumulation on fixed tissue sections from organotypic retinal explant cultures derived from *rd1*Cngb1^−/−^* mice. These cultures were then exposed to either D-cis-diltiazem or Olaparib, with untreated *wt* explants serving as additional controls.

In *wt* retina, both the numbers of ONL cells displaying PARP activity and PAR accumulation were much lower (Figure 5a,b, *cf*. Figure 2 and Appendix A
Figure A1) when compared to *rd1*Cngb1^−/−^* double-mutant retina (Figure 5c,d; quantification in Figure 5i,j). Nevertheless, *rd1*Cngb1^−/−^* double-mutant retina displayed fewer PARP activity and PAR-positive cells in the ONL when compared with *rd1* single mutants (*cf*. Figure 5 with Figure 2), indicating that CNG channels might be related to the regulation of PARP activity. When treated with D-cis-diltiazem, the percentages of photoreceptors showing PARP activity and PAR accumulation were significantly decreased in *rd1*Cngb1^−/−^* retina (Figure 5e,f; PARP activity assay: *p* = 0.0011; PAR staining: *p* < 0.0001). When treated with Olaparib, *rd1*Cngb1^−/−^* retina showed a similar marked reduction of PARP activity and PAR accumulation (Figure 5g,h; *p* < 0.0001). This data again hinted at a relationship between VGCC opening and PARP activity.

### 3.6. Effect of D-cis-diltiazem and Olaparib on rd1*Cngb1^−/−^ Photoreceptor Degeneration

To evaluate the effect of VGCC and PARP inhibition on photoreceptor degeneration, we performed TUNEL staining to label cell death on organotypic retinal explant cultures derived from *rd1*Cngb1^−/−^* mice and treated these with either D-cis-diltiazem or Olaparib.

Remarkably, D-cis-diltiazem significantly reduced the percentage of TUNEL-positive cells in the ONL of *rd1*Cngb1^−/−^* retina when compared to the untreated control (*p* = 0.0134, Figure 6b,c, quantification in e). In contrast, Olaparib did not show a similar effect on cell death in the *rd1*Cngb1^−/−^* retina (Figure 6b,d). These results indicated that, in the absence of rod CNG channel function, the cell death of photoreceptors depended on VGCC, but not on PARP, activity.

## 4. Discussion

In IRDs, excessive activation of PARP and calpain is closely related to the execution of cGMP-induced cell death [5], yet it has been unclear whether these two enzymes act independently or in concert within the same cell death pathway. Our present study confirms that PARP and calpain take part in the same pathway and that PARP activity occurs upstream of calpain activity. We also show that the two major Ca^2+^ sources in photoreceptors, the CNG channel and VGCC, contribute to PARP activation.

### 4.1. Calpain Activation Occurs Downstream of PARP

Calpains belong to a family of Ca^2+^-dependent thiol proteases of which fifteen members have been identified to date [20]. The best-characterized calpains in the central nervous system are two distinct heterodimeric subtypes: calpain-1 and calpain-2 [38], also known as μ-calpain and m-calpain, since they are activated by 3–50-μM Ca^2+^, (i.e., micromolar Ca^2+^) and 0.4–0.8-mM Ca^2+^ (i.e., millimolar Ca^2+^), respectively [39]. Calpain-1 and calpain-2 are thought to play opposing roles in neurodegeneration, with the activation of calpain-1 counteracting degeneration of neurons and calpain-2 promoting it [25,40,41]. Typically, these two calpains are not active at the same time and may even inactivate each other [24,42]. Our immunostaining for activated calpain-2 suggests that, in degenerating photoreceptors, most of the signal detected by the in situ calpain activity assay stems from calpain-2 activity. From non-photoreceptor cells, activated calpains are known to degrade substrate proteins, such as α-spectrin, Bcl-2 family members, RIP kinase, apoptosis-inducing factor (AIF), and PARP-1 [28,29,38,43,44]. Calpastatin as the endogenous inhibitor of calpains, inhibits predominantly calpain-1 and calpain-2 but also reduces the activity of calpain-8 and calpain-9 [45]. Accordingly, the application of calpastatin in our experiments decreased both the calpain activity and calpain-2 immunosignal and delayed photoreceptor degeneration in *rd1* retinal explants, as demonstrated by the TUNEL assay. This data is consistent with previous research [26] and highlights the importance of calpain-dependent proteolysis for photoreceptor degeneration.

Based on previous reports that found calpain to be able to cleave PARP, our initial hypothesis was that calpain might act upstream of PARP during photoreceptor degeneration [28,29]. However, treatment of retinal explants with calpastatin had no effect on PARP activity or the accumulation of PAR. Yet, the other way around, the PARP inhibitor Olaparib strongly decreased calpain activity. Together, this provided a strong indication that activation of PARP occurred upstream of calpain.

### 4.2. VGCC and CNG Channel Contribute to Calpain Activation

When compared with the calpastatin treatment, D-cis-diltiazem had a very similar effect on calpain activity in *rd1* retina, suggesting that the L-type calcium channel found in the cell bodies and at the synaptic terminals of rods was involved in providing the Ca^2+^ required for calpain activation. Even in the absence of *Cngb1* expression, photoreceptor calpain activity can be observed, further supporting the role of VGCCs in calpain activation [46]. D-cis-diltiazem administration also significantly reduced *rd1* PARP activity. However, D-cis-diltiazem did not significantly reduce the TUNEL signal in *rd1* retinal explants, in line with previous studies that found that pharmacological inhibition or genetic inactivation of VGCCs had only a short-term effect, if at all, on delaying *rd1* photoreceptor degeneration [27,47,48]. Since D-cis-diltiazem may also produce oxidative stress in depolarized rods, this may have offset the beneficial effects of the reduction of PARP and calpain activity [36].

The major Ca^2+^ source in rod photoreceptor outer segments is the rod CNG channel, a heterotetrametric cGMP-gated cation channel assembled by three *Cnga1* and one *Cngb1* subunits [49]. Loss of the *Cngb1* subunit leads to degradation of the *Cnga1* subunit and, essentially, loss of the rod CNG channel, thus eliminating the cGMP-gated dark current and preventing the rods from responding to light by hyperpolarization [49]. In the *rd1* model, the aberrantly high levels of cGMP lead to over-activation of the CNG channels and, thus, to increased Na^+^- and Ca^2+^- influx and persistent depolarization [7]. Paradoxically, in the *Cngb1* knockout model, where rods lose the ability of light-dependent hyperpolarization, the resting membrane potential remains essentially unchanged, and the cells are constantly depolarized [46]. In *rd1* rod photoreceptors, the strong Na^+^- and Ca^2+^- influx that the CNG channel mediates in its open state needs to be counterbalanced, a function that is fulfilled by the ATP-driven Na^+^/K^+^ exchanger (NKX). Importantly, NKX activity alone represents at least 50% of the total ATP expenditure of a photoreceptor, thereby linking CNG channel activity to energy metabolism [50,51].

Genetic deletion of the *Cngb1* subunit of the rod CNG channel affords robust *rd1* photoreceptor neuroprotection [22], even though genetic deletion of *Cngb1* or the L-type channel *Cacna1f* [52] alone led to photoreceptor degeneration, albeit with slow rates of progression. Moreover, the combined inhibition of VGCC and the CNG channel could interrupt Ca^2+^ homeostasis and cause severe photoreceptor degeneration [53]. On the other hand, when we used D-cis-diltiazem on *rd1*Cngb1^−/−^* mice, photoreceptor cell death was reduced, suggesting that D-cis-diltiazem could potentially exert therapeutic effects in the absence of CNG channel function. Although D-cis-diltiazem has been reported to partially inhibit CNG channels in rod outer segments [54,55], our data indirectly rule out that it had an effect on rod degeneration in *rd1* retinal explants.

### 4.3. PARP Regulates Calpain via a Pathway That Depends on CNG Channel Function

An abnormal activation of PARP in degenerating photoreceptors is well-documented [5,6,56,57], and accordingly, the PARP inhibitor Olaparib decreased photoreceptor cell death, consistent with previous studies [13,58,59,60,61]. PARP inhibitors are primarily used for cancer therapy due to their ability to prevent DNA repair, and several PARP inhibitors are being tested clinically or have already been approved for clinical use [62]. Notably, Olaparib became the first PARP inhibitor to be approved by the FDA to treat metastatic breast cancer in January 2018 [63]. Currently, Olaparib is the most specific PARP inhibitor available and has no binding to 392 unique human kinases [64]. Among the known off targets is inosine monophosphate dehydrogenase 2 (IMPDH2) [65], which shares 84% sequence homology with IMPDH1, an isozyme that is strongly expressed in the retina [66]. Since IMPDH1 catalyzes the rate-limiting step of GTP production, the substrate employed by retinal guanylate cyclase (retGC) for cGMP synthesis [14], an off-target inhibition of IMPDH1 by Olaparib could potentially reduce cGMP production [67]. However, our cGMP immunostaining revealed that the number of cGMP-positive cells in the *rd1* retina was not decreased in the Olaparib-treated group when compared to the control, suggesting that interactions of Olaparib with IMPDH1 were not responsible for the effects observed.

PARP activity consumes large amounts of NAD^+^ with important ramifications for cell metabolism. Since the biosynthesis of NAD^+^ occurs directly from nicotinamide mononucleotide (NMN) and ATP via nicotinamide mononucleotide adenylyl transferases (NMNATs), excessive consumption of NAD^+^ can lead to ATP depletion [68]. Additionally, NAD^+^ is a key substrate in the tricarboxylic acid cycle, and its deficiency could result in mitochondrial dysfunction [69], which can further aggravate ATP depletion [70]. Moreover, PARP joins ADP-ribose units from NAD^+^ to form PAR [17], high levels of which may also induce mitochondrial dysfunction [71]. The dysfunction induced by NAD^+^ depletion and/or PAR accumulation may depolarize the mitochondrial membrane potential and open the mitochondrial permeability transition pore [72], which could then release mitochondrial Ca^2+^ to the cytoplasm and activate Ca^2+^-dependent proteases [73,74,75]. Therefore, it appears likely that PARP inhibition, through reduced NAD^+^ consumption and PAR generation, will alleviate mitochondrial stress. This, in turn, should increase the availability of ATP that may be used to, for instance, drive the plasma membrane Ca^2+^- ATPase (PMCA), which, in turn, will reduce the intracellular Ca^2+^- levels and, thus, calpain activity.

In *rd1*Cngb1^−/−^* mice, the lack of CNGB1 strongly reduces the Na^+^- and Ca^2+^- influx [7,76]. In *rd1*Cngb1^−/−^* retina, Olaparib treatment did not reduce the calpain activity, calpain-2 activation, or photoreceptor cell death. This points to a mechanism where CNG channel function is essential for PARP-dependent activation of calpain and the execution of photoreceptor cell death. On the other hand, in *rd1*Cngb1^−/−^* retinal explants, D-cis-diltiazem treatment reduced PARP activity. This suggests that, in the absence of CNG channel function, PARP activity is dependent, at least in part, on VGCC-mediated Ca^2+^- influx. Indeed, PARP activity has been associated with elevated intracellular Ca^2+^- levels [77,78,79], and it is thus conceivable that both CNG channels and/or VGCCs contribute to PARP activation. The exact nature of this detrimental interaction between CNG channels, VGCCs, and PARP remains unclear but could potentially be related to alterations of Na^+^- and Ca^2+^- homeostasis and the increased demands on cellular metabolism that this represents. Moreover, because of the rapid degeneration phenotype in *rd1* mice, the results obtained here relate to retina that is still not fully developed and immature. Future studies may reveal to what extent the results obtained in early postnatal retina can be extended to mature adult retina and, if so, how this may apply to IRD patients. At any rate, in genetically very heterogenous IRDs, these findings, together with the fact that cGMP-induced cell death may show an overlap with PARthanatos [6], highlight PARP as a promising target for the development of mutation-independent therapies.

### 4.4. Connecting VGCCs and CNG Channels with the Activity of PARP and Calpain

In the following we will attempt to summarize the main results of this study and provide an overview of how the various experimental conditions may have affected the metabolism in the photoreceptor outer segment, inner segment, cell body, and nucleus and how, in turn, this may promote photoreceptor survival. As discussed above, high cGMP observed in degenerating *rd1* rod photoreceptors activates the CNG channel in the outer segment, leading to Ca^2+^- influx and depolarization [7], which then can activate VGCCs in the cell body, causing more Ca^2+^- influx (Figure 7). In the cell body, Ca^2+^- extrusion depends largely on the ATP-driven plasma membrane Ca^2+^-ATPase (PMCA) [51]. A lack of ATP in degenerating rods may cause a decrease of PMCA activity and further potentiate the accumulation of intracellular Ca^2+^. In either case, high Ca^2+^ levels may activate calpain-type proteases [73], notably calpain-2, which is considered to promote neuronal degeneration [38]. Independent of CNG channels, high cGMP levels can also activate cGMP-dependent protein kinase (PKG), which may be associated with histone deacetylase (HDAC) activity in the nucleus, leading to chromatin condensation and DNA damage [6]. This, in turn, may trigger the over-activation of PARP [6] and cause the execution of a PARthanatos-related form of cell death [19,80] (Figure 7).

Treatment with calpain inhibitors, such as calpastatin, may decrease proteolytic damage and thereby prolong photoreceptor survival. Further upstream, blocking VGCCs with D-cis-diltiazem will reduce the Ca^2+^-levels in the photoreceptor cell body and prevent calpain activation. Moreover, since VGCCs appear to be involved in PARP activation, reduced PARP activity and PAR levels likely alleviate mitochondrial stress, allowing the cell to maintain ATP production. This becomes evident through the use of the PARP inhibitor Olaparib, which decreases PAR generation. This, in turn, may preserve mitochondrial function and intracellular ATP levels, enabling PMCA to extrude Ca^2+^ and keep the calpain activity low (Figure 7).

## 5. Conclusions

In IRDs, the activities of calpain and PARP are both closely associated with photoreceptor degeneration, and PARP is a known target for calpain-dependent cleavage. However, here, we demonstrate that, in cGMP-induced photoreceptor degeneration, PARP regulates calpain activity, likely in an indirect fashion, via NAD^+^/ATP depletion or PAR-induced mitochondrial dysfunction. In addition, PARP activity is likely to be controlled by the activities of the VGCC and CNG channels. Overall, these results suggest PARP as a particularly attractive target for future therapeutic interventions in IRDs. The availability of a number of clinically tested PARP inhibitors [81,82] further enhances the perspectives for clinical translation.

## Figures and Tables

**Figure 1 biomolecules-12-00455-f001:**
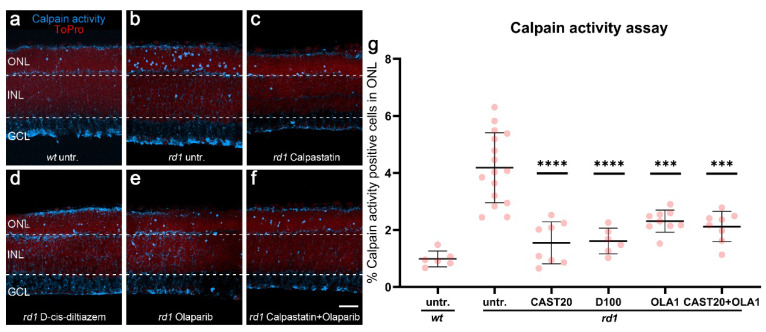
Effects of calpastatin, D-cis-diltiazem, Olaparib, and combination treatments on calpain activity. The calpain activity assay (blue) was performed on unfixed *wt* (**a**) and *rd1* retinal cross-sections. ToPro (red) was used as nuclear counterstaining. Untreated *rd1* retina (untr.; **b**) was compared to retina treated with either calpastatin (**c**), D-cis-diltiazem (**d**), Olaparib (**e**), or calpastatin and Olaparib combined (**f**). The scatter plots show the percentages of outer nuclear layer (ONL) cells positive for calpain activity (**g**) in *wt* and *rd1* retina. Statistical significance was assessed using one-way ANOVA and Tukey’s multiple comparison post hoc testing performed between the control (*rd1* untreated) and 20-μM calpastatin (CAST20), 100-μM D-cis-diltiazem (D100), 1-μM Olaparib (OLA1), and 20-μM calpastatin combined with 1-μM Olaparib (CAST20+OLA1). All treatments reduced the calpain activity in *rd1* ONL; however, there was no added synergistic benefit from the CAST20+OLA1 combination. Untr. *wt*: 6 explants from 3 different mice; untr. *rd1*: 16/16; CAST20 *rd1*: 8/8; D100 *rd1*: 6/6; OLA1 *rd1*: 9/9; CAST20+OLA1 *rd1*: 8/8; error bars represent SD; *** = *p* ≤ 0.001 and **** = *p* ≤ 0.0001. INL = inner nuclear layer, GCL = ganglion cell layer. Scale bar = 50 µm.

**Figure 2 biomolecules-12-00455-f002:**
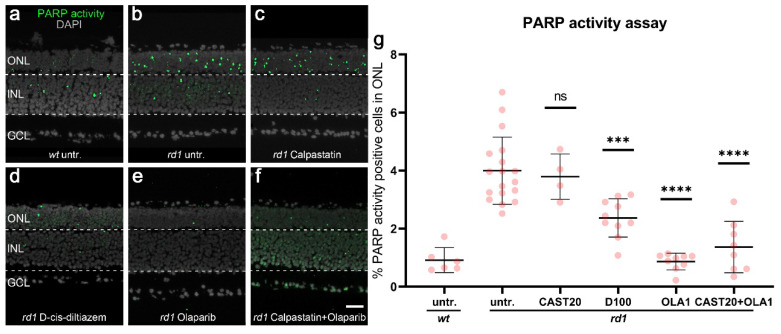
Effects of calpastatin, D-cis-diltiazem, Olaparib, and combination treatments on PARP activity. PARP activity assay (green) in *wt* and *rd1* retina. DAPI (grey) was used as nuclear counterstaining. Untreated *wt* (untr.; **a**) was compared to untreated *rd1* retina (**b**) and retinae treated with either calpastatin (**c**), D-cis-diltiazem (**d**), Olaparib (**e**), or calpastatin and Olaparib combined (**f**). The scatter plots show the percentages of the outer nuclear layer (ONL) cells positive for PARP activity (**g**) in *wt* and *rd1* retina. Statistical significance was assessed using one-way ANOVA and Tukey’s multiple comparison post hoc testing performed between the control (*rd1* untreated) and 20-μM calpastatin (CAST20), 100-μM D-cis-diltiazem (D100), 1-μM Olaparib (OLA1), and 20-μM calpastatin combined with 1-μM Olaparib (CAST20+OLA1). Calpastatin did not reduce the PARP activity, while D-cis-diltiazem and Olaparib did. Untr. *wt*: 6 explants from 3 different mice; untr. *rd1*: 18/18; CAST20 *rd1*: 4/4; D100 *rd1*: 10/10; OLA1 *rd1*: 9/9; CAST20+OLA1 *rd1*: 8/8; error bars represent SD; ns = *p* > 0.05, *** = *p* ≤ 0.001, and **** = *p* ≤ 0.0001. INL = inner nuclear layer, GCL = ganglion cell layer. Scale bar = 50 µm.

**Figure 3 biomolecules-12-00455-f003:**
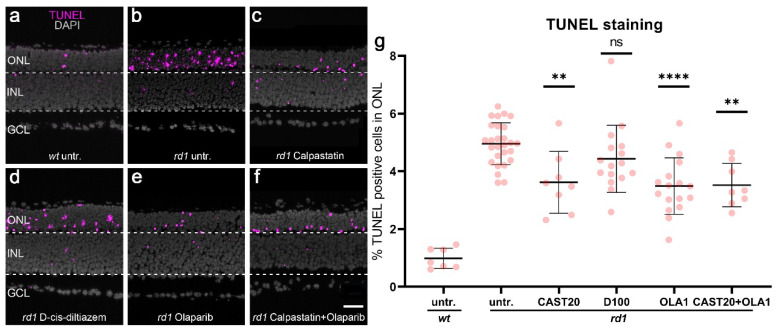
Effects of calpastatin, D-cis-diltiazem, Olaparib, and combination treatments on *rd1* retinal cell viability. The TUNEL assay labeled dying cells (magenta) in *wt* and *rd1* retinal explant cultures. DAPI (grey) was used as a nuclear counterstain. Untreated *wt* (**a**) and *rd1* control retina (untr.; **b**) were compared to retina treated with either 20-µM calpastatin (CAST20, **c**), 100-µM D-cis-diltiazem (D100, **d**), 1-µM Olaparib (OLA1, **e**), or 20-µM calpastatin combined with 1-µM Olaparib (CAST20+OLA1, **f**). Note the large numbers of dying cells in the *rd1* outer nuclear layer (ONL). The scatter plot (**g**) shows the percentage of TUNEL-positive cells in the ONL. Statistical significance was assessed using one-way ANOVA and Tukey’s multiple comparison post hoc testing performed between the control (*rd1* untreated) and 20-μM calpastatin (CAST20), 100-μM D-cis-diltiazem (D100), 1-μM Olaparib (OLA1), and 20-μM calpastatin combined with 1-μM Olaparib (CAST20+OLA1). Except for D-cis-diltiazem, all treatments decreased *rd1* retinal degeneration. The combination treatment CAST20+OLA1 did not improve the therapeutic effect seen with the respective single treatments. Untr. *wt*: 7 explants from 4 different mice; untr. *rd1*: 27/27; CAST20 *rd1*: 8/8; D100 *rd1*: 16/16; OLA1 *rd1*: 17/17; CAST20+OLA1 *rd1*: 8/8; error bars represent SD; ns = *p* > 0.05, ** = *p* ≤ 0.01, and **** = *p* ≤ 0.0001. ONL = outer nuclear layer, INL = inner nuclear layer, GCL = ganglion cell layer. Scale bar = 50 µm.

**Figure 4 biomolecules-12-00455-f004:**
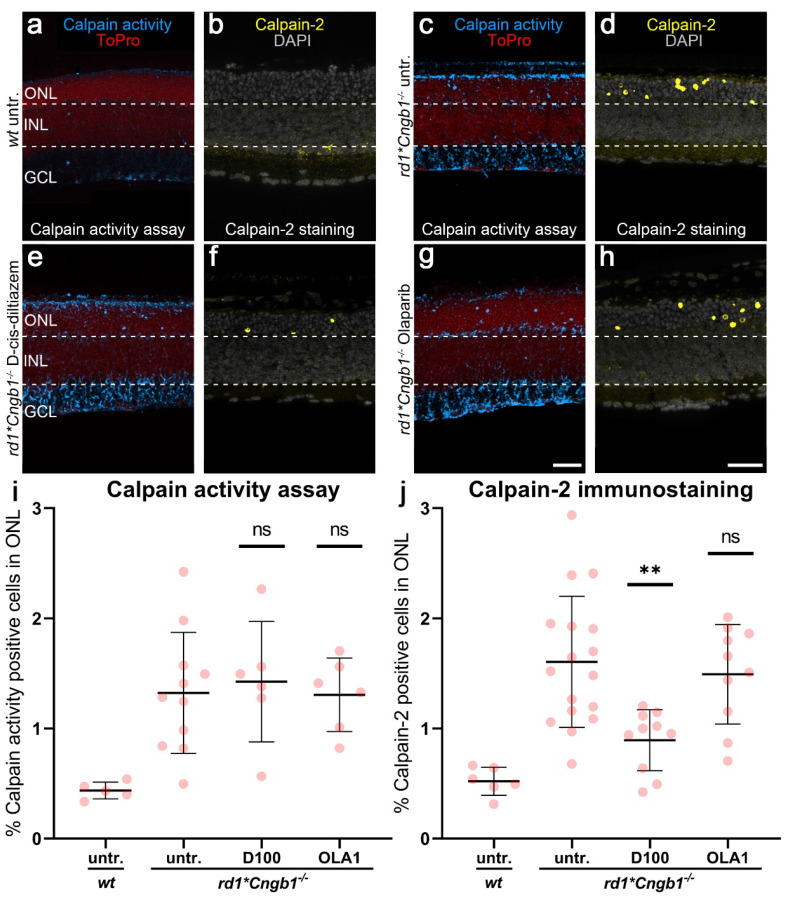
Effects of D-cis-diltiazem and Olaparib on calpain activity in *rd1*Cngb1^−/−^* retina. The calpain activity assay (blue) and an immunostaining for activated calpain-2 (yellow) were performed on *wt* (**a**,**b**) and *rd1*Cngb1^−/−^* retina. DAPI (grey) was used as nuclear counterstaining. Untreated *rd1*Cngb1^−/−^* retina (untr.; **c**,**d**) was compared to retina treated with D-cis-diltiazem (**e**,**f**) or Olaparib (**g**,**h**). The scatter plots show the percentages of ONL-positive cells for calpain activity (**i**) and activated calpain-2 (**j**) in the *wt* and treated *rd1*Cngb1^−/−^* retina compared with the *rd1*Cngb1^−/−^* control (untr.). Statistical significance was assessed using one-way ANOVA and Tukey’s multiple comparison post hoc testing performed between the control (*rd1*Cngb1^−/−^* untreated), 100-μM D-cis-diltiazem (D100), and 1-μM Olaparib (OLA1). In *rd1*Cngb1^−/−^*, only D-cis-diltiazem reduced the cells positive for activated calpain-2. In the calpain activity assay, untr. *wt*: 5 explants from 3 different mice; untr. *rd1***Cngb1*^−/−^: 11/11; D100 *rd1***Cngb1*^−/−^: 6/6; OLA1 *rd1***Cngb1*^−/−^: 6/6; in calpain-2 immunostaining, untr. *wt*: 6/3; untr. *rd1***Cngb1*^−/−^: 17/17; D100 *rd1***Cngb1*^−/−^: 10/10; OLA1 *rd1***Cngb1*^−/−^: 10/10; error bars represent SD; ns = *p* > 0.05 and ** = *p* ≤ 0.01. ToPro (red) and ONL = outer nuclear layer, INL = inner nuclear layer, GCL = ganglion cell layer. Scale bar = 50 µm.

**Figure 5 biomolecules-12-00455-f005:**
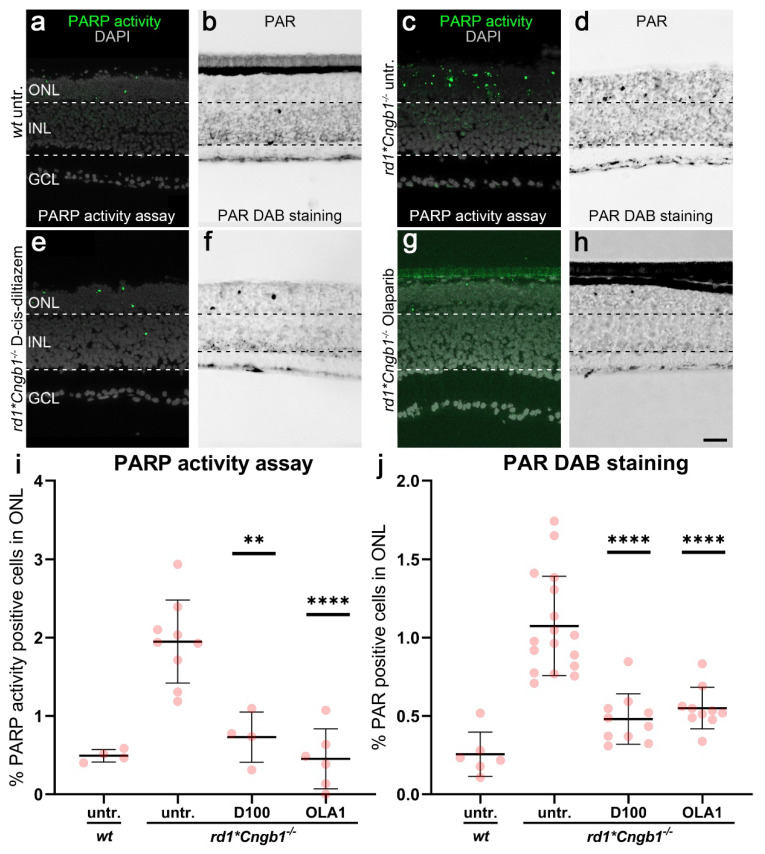
Effects of D-cis-diltiazem and Olaparib on PARP activity and PAR accumulation in *rd1*Cngb1^−/−^* double-mutant retina. The PARP activity assay (green) and immunostaining for PAR (black) were performed on *wt* (**a**,**b**) and *rd1*Cngb1^−/−^* retina (**c**–**h**). DAPI (grey) was used as nuclear counterstaining. Untreated *rd1*Cngb1^−/−^* retina (untr.; **c**,**d**) was compared to retina treated with D-cis-diltiazem (**e**,**f**) or Olaparib (**g**,**h**). The scatter plots show the percentages of outer nuclear layer (ONL) cells positive for PARP activity (**i**) and PAR (**j**) in *wt* and treated *rd1*Cngb1^−/−^* retina compared to the *rd1*Cngb1^−/−^* control (untr.). Statistical significance was assessed using one-way ANOVA and Tukey’s multiple comparison post hoc testing performed between the control (*rd1*Cngb1^−/−^* untreated) and 100-μM D-cis-diltiazem (D100) or 1-μM Olaparib (OLA1). D-cis-diltiazem strongly decreased the PARP activity and PAR. In the PARP activity assay, untr. *wt*: 4 explants from 2 different mice; untr. *rd1***Cngb1*^−/−^: 9/9; D100 *rd1***Cngb1*^−/−^: 4/4; OLA1 *rd1***Cngb1*^−/−^: 6/6. In PAR DAB staining, untr. *wt*: 6/3; untr. *rd1***Cngb1*^−/−^: 17/17; D100 *rd1***Cngb1*^−/−^: 10/10; OLA1 *rd1***Cngb1*^−/−^: 10/10; error bars represent SD; ** = *p* ≤ 0.01 and **** = *p* ≤ 0.0001. INL = inner nuclear layer, GCL = ganglion cell layer. Scale bar = 50 µm.

**Figure 6 biomolecules-12-00455-f006:**
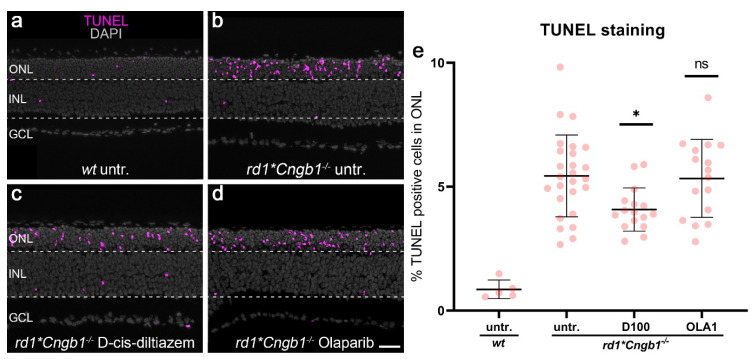
Effects of D-cis-diltiazem and Olaparib on *rd1*Cngb1^−/−^* retinal cell viability. The TUNEL assay labeled dying cells (magenta) in wild-type (*wt*) and *rd1*Cngb1^−/−^* retinal explant cultures. DAPI (grey) was used as a nuclear counterstain. (**a**) In *wt* retina, only a small fraction of cells in the outer nuclear layer (ONL) were TUNEL-positive. (**b**) Untreated (untr.) *rd1*Cngb1^−/−^* double-mutant retina was compared to retina treated with either 100-µM D-cis-diltiazem (D100, (**c**) or 1-µM Olaparib (OLA1, (**d**,**e**) The scatter plot shows the percentage of TUNEL-positive cells. Statistical significance was assessed using one-way ANOVA and Tukey’s multiple comparison post hoc testing performed between the control (*rd1*Cngb1^−/−^* untreated) and 20-μM calpastatin (CAST20), 100-μM D-cis-diltiazem (D100), 1-μM Olaparib (OLA1), and 20-μM calpastatin combined with 1-μM Olaparib (CAST20+OLA1). Only D-cis-diltiazem alleviated the *rd1*Cngb1^−/−^* retinal degeneration. Untr. *wt*: 5 explants from 3 different mice; untr. *rd1***Cngb1*^−/−^: 26/26; D100 *rd1***Cngb1*^−/−^: 16/16; OLA1 *rd1***Cngb1*^−/−^: 16/16; error bars represent SD; ns = *p* > 0.05 and * = *p* ≤ 0.05. INL = inner nuclear layer, GCL = ganglion cell layer. Scale bar = 50 µm.

**Figure 7 biomolecules-12-00455-f007:**
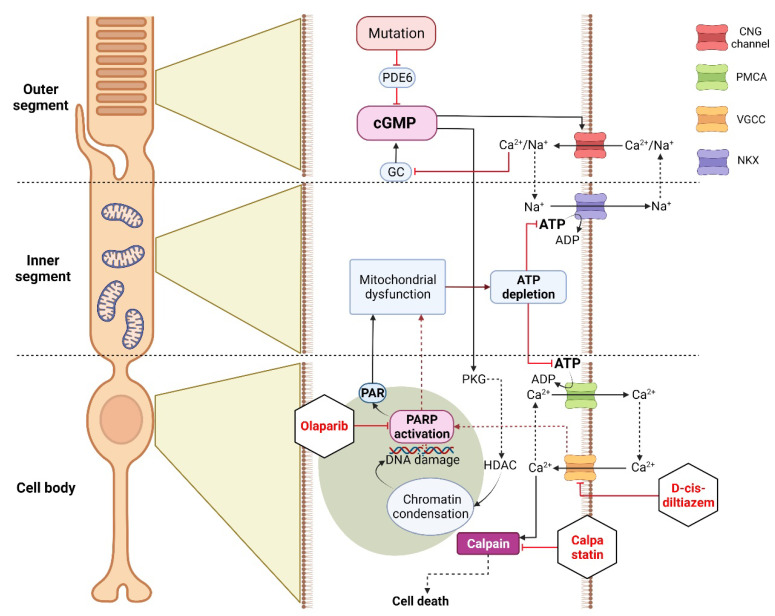
Differential effects of experimental conditions on cGMP-dependent cell death in *rd1* photoreceptors. The mutation-induced cGMP accumulation activates cyclic nucleotide-gated (CNG) channels in the outer segment, leading to Na^+^-and Ca^2+^-influx and photoreceptor depolarization. This leads to opening of voltage-gated Ca^2+^-channels (VGCCs) in the cell body, causing further Ca^2+^- influx. In the cell body, high Ca^2+^ levels may activate calpain if not controlled by ATP-dependent plasma membrane Ca^2+^-ATPase (PMCA). In addition, cGMP-dependent activation of protein kinase G (PKG) has been associated with histone-deacetylase (HDAC) activity, causing chromatin condensation and DNA breaks, which may trigger PARP activation. Excessive consumption of NAD^+^ by PARP and the production of PAR may cause mitochondrial dysfunction, leading to ATP shortage. Calpastatin treatment blocks calpain activation, decreasing proteolytic damage to the cell, even in the presence of CNG channel/VGCC-mediated Ca^2+^-influx. D-cis-diltiazem inhibits VGCCs in the cell body, reducing intracellular Ca^2+^-levels and calpain activity. Moreover, VGCCs could be involved in PARP activation, even though D-cis-diltiazem fails to delay *rd1* rod degeneration. Olaparib blocks PARP activity, decreasing NAD^+^ consumption and PAR generation. This may preserve mitochondrial function and intracellular ATP levels, allowing PMCA to extrude Ca^2+^ and keeping calpain activity low.

## Data Availability

Not applicable.

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
