# Peer review of "Inherited Retinal Degeneration: PARP-Dependent Activation of Calpain Requires CNG Channel Activity"

_biomolecules, 2022, doi:10.3390/biom12030455_

Round 1

Reviewer 1 Report

Authors show that inhibition of either calpain or PARP reduces photoreceptor cell death in Pde6b-mutant rd1 mouse retinal explants. Inhibition of PARP reduced calpain activity, whereas calpain did not alter PARP activity, and no synergistic effect on prevention of cell death was observed. Suggested is that both enzymes act in the same degenerative pathway triggered by high levels of cGMP, with PARP acting upstream of calpain (in this retinal degeneration model). In rd1*Cngb1-/- double-mutant mouse retinal explants which show slower retinal degeneration than in rd1, PARP inhibition did not delay photoreceptor cell death whereas the inhibition of voltage-gated Ca2+-channels (VGCCs) did. PARP activity in turn might be controlled by VGCC. Authors suggest that PARP is a potential target for therapeutic clinical intervention.

Major

1) Authors studied the effects of inhibitors on developing retinal explants (cultured from P5 up to P11 in rd1 mouse retina; cultured from P5 up to P17 in rd1Cngb1 mouse retina). Authors suggest that PARP is a potential target for therapeutic intervention. Whereas the observations are of interest, the authors should discuss the relevance of these effects observed in developing retinal explants with the phenotypes in adult patients. What is the biological relevance, especially for using PARP as potential target for therapy on patients?

2) On the statistical analysis, please report the number of biological replicates. For example in the legend to Fig 6, the number of retinal explants is indicated as n=16 to 26, but from how many mice obtained? It is highly likely that there is variation in the retinal phenotypes between mice. And when the mice are not on exactly the same genetic background than there might be high variation. Legend as well the section on statistical analysis in Materials and Methods is not sufficiently clear on this. Analysis should be on at least 3 preferably 5 independent biological replicates (different mice) for such comparative analyses. From the data presented I did get the impression that it might be from many retinal explants obtained from a limited number of mouse retinas. There might be variation in expression of the PARP, calpain(s), and other enzymes in the retinal explants of different mutant mice. Please clarify.

Author Response

  • Authors studied the effects of inhibitors on developing retinal explants (cultured from P5 up to P11 in rd1 mouse retina; cultured from P5 up to P17 in rd1*Cngb1 mouse retina). Authors suggest that PARP is a potential target for therapeutic intervention. Whereas the observations are of interest, the authors should discuss the relevance of these effects observed in developing retinal explants with the phenotypes in adult patients. What is the biological relevance, especially for using PARP as potential target for therapy on patients?

Answer: Thank you for your advice. In response to this question we have added a comment to the introduction on the prevalence of human PDE6 gene mutations (page2, lines 54-56) and added a further comment in the discussion on how our data may apply to adult retina in rodents and man (page 13, lines 500-504).

  • On the statistical analysis, please report the number of biological replicates. For example, in the legend to Fig 6, the number of retinal explants is indicated as n=16 to 26, but from how many mice obtained? It is highly likely that there is variation in the retinal phenotypes between mice. And when the mice are not on exactly the same genetic background than there might be high variation. Legend as well the section on statistical analysis in Materials and Methods is not sufficiently clear on this. Analysis should be on at least 3 preferably 5 independent biological replicates (different mice) for such comparative analyses. From the data presented I did get the impression that it might be from many retinal explants obtained from a limited number of mouse retinas. There might be variation in expression of the PARP, calpain(s), and other enzymes in the retinal explants of different mutant mice. Please clarify.

Answer: We thank the reviewer for this question. We have now added the number of individual retinal explants and the number of different mice that we have used in each figure legend. In our figures, each dot represents an individual retinal explant culture obtained from a different animal (i.e., the two retinae from one animal were split across different experimental groups). We have also now inserted a corresponding clarification in the Materials and Methods section (page 3, lines 107-109).

Reviewer 2 Report

The paper entitled "Inherited retinal degeneration: PARP-dependent activation of 2 calpain requires CNG channel activity " and submitted for publication in the journal biomolecules is an interesting, quality and well written paper.

Despite this quality, I have two questions that I would like to see clarified by the authors:
1) The images that the authors present of the retinal sections do not allow me to observe, with the necessary quality, the layers of the retina. Due to this low quality, they also do not allow me to understand how the authors counted the total number of cells present in the various cell layers. This is important because the authors express their results as % of the total number of cells (for example: % calpain activity positive cells in ONL). This may be due to the small size of the images, so I would ask the authors to detail how they performed the cell counting (manual or automatic?) and to change the images to larger images that allow better observation of the various retinal layers. As an option, I ask that you make the original images available to the reviewers, they are large enough for careful analysis.
2) The authors present results that are not in agreement with others they have recently published (Das, S., Popp, V., Power, M. et al. Redefining the role of Ca2+-permeable channels in photoreceptor degeneration using diltiazem. Cell Death Dis 13, 47 (2022). https://doi.org/10.1038/s41419-021-04482-1). In this work, which is not referenced in the manuscript, it is shown that the application of D-cis-diltiazem 100uM to the retinas of rd1 animals did not significantly reduce the increase in calpain activity and, in addition, caused a slight increase in cell death, as measured by TUNEL. In the manuscript now submitted the authors observe, in the same experimental model, that the application of the VSCC inhibitor induces a significant reduction in calpain activity in the ONL and had a slight protective effect on cell death, translated into a reduction in the number of TUNEL+ cells in the ONL. I think it is necessary that the authors discuss these contradictions, especially considering the fact that some of the authors participated in both papers.

A minor remark concerns the fact that the authors indicate in the figures GCs (ganglion cells). The authors should caption the figure with GCL (ganglion cell layer) because the nuclear markers they used, such as DAPI, mark other cells present in that layer besides the ganglion cells. 

Author Response

  • The images that the authors present of the retinal sections do not allow me to observe, with the necessary quality, the layers of the retina. Due to this low quality, they also do not allow me to understand how the authors counted the total number of cells present in the various cell layers. This is important because the authors express their results as % of the total number of cells (for example: % calpain activity positive cells in ONL). This may be due to the small size of the images, so I would ask the authors to detail how they performed the cell counting (manual or automatic?) and to change the images to larger images that allow better observation of the various retinal layers. As an option, I ask that you make the original images available to the reviewers, they are large enough for careful analysis.

Answer: We thank the reviewer for this advice. We have clarified the methodology of positive cell quantification in section 2.8 (page 5, lines 181-186), and we furthermore now supply the original, full-resolution images for review purposes.

  • The authors present results that are not in agreement with others they have recently published (Das, S., Popp, V., Power, M. et al. Redefining the role of Ca2+-permeable channels in photoreceptor degeneration using diltiazem. Cell Death Dis 13, 47 (2022). https://doi.org/10.1038/s41419-021-04482-1). In this work, which is not referenced in the manuscript, it is shown that the application of D-cis-diltiazem 100uM to the retinas of rd1 animals did not significantly reduce the increase in calpain activity and, in addition, caused a slight increase in cell death, as measured by TUNEL. In the manuscript now submitted the authors observe, in the same experimental model, that the application of the VSCC inhibitor induces a significant reduction in calpain activity in the ONL and had a slight protective effect on cell death, translated into a reduction in the number of TUNEL+ cells in the ONL. I think it is necessary that the authors discuss these contradictions, especially considering the fact that some of the authors participated in both papers.

Answer: We thank the reviewer for this pertinent observation. We have in fact cited the Das et al., 2022 article as reference #53 in the discussion (page 13, line 450).

In terms of the calpain activity assay, there are differences between that article and our present study. These concern the use of different calpain substrate concentrations. Although our earlier article (Das et al., 2022) was published only recently, the calpain activity assays for that article were performed more than 2 years ago. At that time, we were using 5 µM tBOC-Leu-Met-CMAC (Thermofisher Scientific, A6520), while in the present study we increased the concentration of the substrate to 25 µM (see Materials and Methods, page 3, lines 135-140). This technical improvement leads to a higher detection rate for calpain activity positive cells across all samples and experimental conditions.

Moreover, and perhaps more importantly, the earlier study used only 3-4 independent retinal explants per experimental group. While this was sufficient to determine the very marked treatment effects of L-cis-diltiazem, the Das et al., study was underpowered to resolve the relatively weaker differences caused by the D-cis-diltiazem treatment. In the present study, we have made a substantial effort to remedy this and now include many more independent retinal explant samples and animals (6-16 for calpain activity assay, 10-15 for activated calpain-2).

As far as the TUNEL staining is concerned, in the previous Das et al., study there were no significant differences between untreated and D-cis-diltiazem treated rd1 retina. The same is true for our current study. An interesting effect in our present study, is that in rd1*Cngb1-/- double-mutant retina, in contrast to what is seen in rd1 retina, D-cis-diltiazem produces a significant reduction of TUNEL staining (as well as for PARP and calpain activity).   

A minor remark concerns the fact that the authors indicate in the figures GCs (ganglion cells). The authors should caption the figure with GCL (ganglion cell layer) because the nuclear markers they used, such as DAPI, mark other cells present in that layer besides the ganglion cells. 

Answer: Thank you for your suggestion. We have changed the caption “GCs” to now read “GCL” in all figures and legends.

Round 2

Reviewer 2 Report

The authors responded satisfactorily to the questions raised and made the suggested changes to the manuscript, so I believe that now, in its current form, the manuscript meets the necessary conditions for publication.